# Independent acoustic variation of the higher- and lower-frequency components of biphonic calls can facilitate call recognition and social affiliation in killer whales

**Olga A. Filatova** *

Department of Vertebrate Zoology, Faculty of Biology, Lomonosov Moscow State University, Moscow, Russia

* alazorro@gmail.com

## Abstract

Each resident-type (R-type) killer whale pod has a set of stereotyped calls that are culturally transmitted from mother to offspring. The functions of particular call types are not yet clearly understood, but it is believed that calls with two independently modulated frequency components (biphonic calls) play an important role in pod communication and cohesion at long ranges. In this study we examined the possible functions of biphonic calls in R-type killer whales. First, we tested the hypothesis that the additional component enhances the potential of a call to identify the family affiliation. We found that the similarity patterns of the lower- and higher frequency components across the families were largely unrelated. Calls were classified more accurately to their respective family when both lower- and higher-frequency components were considered. Second, we tested the long-range detectability of the lower- and higher-frequency components. After adjusting the received levels by the killer whale hearing sensitivity to different frequency ranges, the sensation level of the higher-frequency component was higher than the amplitude of the lower-frequency component. Our results suggest that the higher-frequency component of killer whale biphonic calls varies independently of the lower-frequency component, which enhances the efficiency of these calls as family markers. The acoustic variation of the higher-frequency component allows the recognition of family identity of a caller even if the shape of the lower-frequency component accidentally becomes similar in unrelated families. The higher-frequency component can also facilitate family recognition when the lower-frequency component is masked by low-frequency noise.

## Introduction

Killer whales are among the few mammalian species that possess vocal learning abilities [1]. Each resident-type (R-type) killer whale pod has a unique call repertoire–vocal dialect consisting from a set of stereotyped calls that calves learn from their mothers and other matrilineal

**Data Availability Statement:** All relevant data are within the manuscript and its Supporting Information files.

**Funding:** This work was supported by Russian Science Foundation (http://www.rscf.ru/) funding to OAF, grant number 19-14-00037. This research received no additional funding from any public, commercial or not-for-profit sectors. The funder had no role in study design, data collection and analysis, decision to publish, or preparation of the manuscript.

**Competing interests:** The author has declared that no competing interests exist.

relatives [2]. The dialects, being patterns of socially learned behavioral phenotypes, therefore represent a form of animal culture [3].

Cultural traditions have been described in many animal species [4]. Social transmission of behavioral innovations is especially beneficial in situations of rapid environmental change, when instinctive behavior that evolves through natural variation and selection is too slow to follow the arising challenges and fast shifts in behavioral adaptations are required. In killer whales, the vocal culture serves a different function. The repertoires change simultaneously with social divergence, and the cultural inheritance of dialects provides a system for recognition of relatedness of the social units [2]. Each R-type killer whale pod shares the same dialect; new pods form gradually through the split of an ancestral pod, and the dialects of the splitting pods slowly diverge in time due to changes in call structure. This provides the complex system of dialects with varying degrees of similarity to each other that more or less reflect the degree of relatedness between the corresponding pods [5, 6].

The functions of particular call types are not yet clearly understood, but it is believed that calls with complex structure consisting of two independently modulated frequency components play important role in pod communication and cohesion at long ranges [7, 8]. These 'biphonic' or 'two-voiced' calls are produced more often during encounters of several pods rather than in a single-pod context [9–11]. Additionally, the differences in directionality of the lower- and higher-frequency components allow a listener to infer the direction of movement of a caller [10].

Biphonation is widespread in mammals and occur in many species as diverse as canids [12–14], primates [15–17] and whales [18, 19]. However, functional interpretations of biphonation are scarce. The proposed functions of this phenomenon include the increase of unpredictability and indication of physical condition [20] and enhancement of individual recognition [20–22]. Volodina [22] reported that the correct individual classification was much higher in dhole (*Cuon alpinus*) calls when using both components of a biphonic call together than when using each of them separately. Among cetaceans, the role of biphonic call in individual recognition has not been tested, though biphonic calls were reported to serve as individual signature sounds in bottlenose dolphins [23, 24].

In R-type killer whales, recognition of a family dialect appears to be at least as important as individual recognition, because a whale stays in its natal family for life, and family affiliation is important for its survival. In contrast to dholes and most other mammals, killer whales possess an ability to change the structure of their calls deliberately in a specific direction [1]. However, the speed of call change is rather slow [25], and optimal complexity of call contours is limited by propagation loss that attenuates fine details of contour structure. In this situation, whales can benefit from ability to change different components of a call independently, which would increase the degree of structural divergence of calls from related families.

Biphonic calls occur in all studied killer whale populations [2, 8, 26–28]. In most populations both in the North Pacific and the North Atlantic, the average frequencies of the higher- and lower-frequency components are more or less similar [29] except for the North Pacific 'transient-type' (T-type) mammal-eating killer whales, which have been shown to be the most genetically divergent group among all studied killer whale populations [30]. The widespread occurrence and homogeneity in frequency of both components across populations suggest that biphonic calls are an essential part of the acoustic repertoire of the killer whale and bear specific functions that are similar in different populations.

In this study we examine the possible functions of biphonic calls in R-type killer whales. First, we test the hypothesis that the additional component enhances the potential of a call to identify the pod and family affiliation. We compare the similarity patterns across the lower- and higher-frequency components and test whether either of the components or both of them

in combination work better to mark the family identity. Second, we test the long-range detectability of the lower- and higher-frequency components by comparing their received levels and adjusting them by killer whale hearing sensitivity to different frequency ranges.

## Material and methods

### Ethics statement

This study was part of the research topic of Lomonosov Moscow State University "Principles of high-frequency and ultrasound communication in mammals". According to the laws of Russian Federation, no permit is required for distant non-invasive research of cetaceans. The field studies did not involve endangered or protected species.

### Data collection

The materials and data for this study were collected as part of the Far East Russia Orca Project (FEROP) in Avacha Gulf, Kamchatka, during the summer months of the years 2012–2019. The underwater sound recordings were made from 4–4.5 m inflatable boats while the engines were turned off, at a sampling frequency of 48 or 96 kHz. For the recording we used Offshore Acoustics hydrophone (nominal sensitivity -154 dB re 1 V/μPa, frequency response curve 6 Hz to 14 kHz 1–3 dB according to manufacturer's specifications) with Zoom H4 and Zoom H6 flash recorders. The photographic identification (photo-ID) method was used to identify individual killer whales and families. To take photographs, we approached the whales to a distance of 20-50m, or moved the boat 200-300m ahead of the animals and off to the side and waited until they passed. Photographs of the left side of individual whales were taken to show the details of dorsal fin and saddle patch, using the technique developed by Bigg et al. [31].

The resident (R-type) killer whales of Eastern Kamchatka, Russia, are known to range along the east coast of Kamchatka peninsula from Avacha Gulf to Karaginsky Gulf and east to the Commander Islands [32]. Whales from this population live in stable social units that include maternal relatives with no dispersal observed [33]. We do not have enough data to reconstruct the full genealogies of these units, and we suspect that in some cases one unit can include more than one matriline, so we use the term "family" to denote these units. Families that share the same vocal dialect are attributed to the same pod, and pods with similar dialects form clans. To date we recognize at least 62 families, which belong to 20 pods in three acoustic clans forming a single community: Avacha clan, K19 clan and K20 clan [33, 34]. Avacha clan, consisting of more than 13 pods and 30 families, is the most common. For this study we used only the families from Avacha clan for which sufficient data were available.

Calls were classified according to the existing catalogue [35] with some additional subtypes identified through our further studies. For this study, we used calls of two most common biphonic types of Avacha clan—K5 and K7 types.

### Similarity of contours of the lower and higher frequency components

For the analysis of contour similarities, I used high-quality calls recorded from particular families when they were alone in the area with no other families nearby. In total I used calls produced by 14 families from five different pods, 10 calls from each type per family (except for K5 call type of Ikar family which had two distinctive subtypes, so we used 10 calls of each subtype from this family, 20 calls from this family in total). To cover intra-group variation, I selected calls from as many independent recording sessions from each family as possible. No sample from any family contained calls from fewer than three independent sessions with that family.

Call contours were extracted using a custom-made MATLAB script for manually tracking frequency contours of each syllable (for the detailed description of the algorithm see [36]. After the operator selected enough points to track all modulations of the frequency contour, the algorithm smoothed and interpolated them to produce a vector of frequency measurements with a sampling interval of 0.01 s. The contours of the lower- and higher-frequency components were extracted independently of each other.

The similarity of contours was measured using dynamic time-warping, which allows limited compression and expansion of a signal's time axis to maximize the frequency overlap with a reference signal. For this study, I used the warping algorithm developed by Deecke and Janik [37]. The percent similarity of contours was calculated by dividing the smaller frequency value by the larger value at each point and multiplying by 100. From the resulting similarity matrix, a cost matrix was constructed that kept a running tab on the similarities of the elements making up the curves while adding up these costs to produce a final number that indicated the percent similarity between the contours. To calculate a distance measure between each pair of calls in our analysis, I subtracted this value from 100%.

In order to estimate the potential ability of killer whales to assess the family identity of calls, I classified the calls using ARTwarp algorithm in MATLAB [37]. The similarity between the extracted contours and the reference contours was calculated using dynamic time-warping. As my aim was to assess the rate of correct classification to existing categories (families), I modified the ARTwarp algorithm used by Deecke and Janik [37] in order to prevent it from creating additional categories and modifying the reference contours. For this, on the first step I calculated a reference contour for each family using the corresponding module of ARTwarp. On the second step, I used model contours as weight matrix and fixed the number of categories, so that all new contours had to be assigned to any of the existing categories, represented by the family reference contours. Then, I calculated the rate of correct classifications for lower- and higher-frequency component contours separately and for the contours that consisted of both components taken together.

## Detectability of the lower- and higher-frequency components over distance

I measured the amplitude in the middle of the fundamental frequency contour of the lower- and higher-frequency components using the rectangle cursor tool in AviSoft SASLab Pro (Fig 1). As the aim of the measurements was to access the difference between the amplitude of the higher- and lower-frequency components, and not the absolute source level values, I did not account for the parameters of the recording system.

The calls were collected for the analysis at different locations within Avacha Gulf. Spectral sound propagation loss can vary between locations due to differences in bathymetry, sediment structure and water properties; some variation is also introduced by differences in depths at which killer whales produce their calls. To address this, we measured a large amount of calls produced in various conditions and in different situations to cover as much natural variation as possible. As killer whales also experience all these propagation effects when listening to the conspecific calls, we assume that our sample set is an adequate representative of what killer whale normally hear in the wild.

I measured the total of 689 K5 calls and 372 K7 calls from 2012–2019. To estimate the average received levels of these call components, I measured all observed calls of these types regardless of their quality and signal-to-noise ratio. If the higher-frequency component was not detectable on the sonogram, the call was not used for the further analysis.

Then, I calculated the difference between the amplitude of the lower- and higher-frequency components as perceived by killer whales. For this, I used killer whale audiogram [38, 39] to

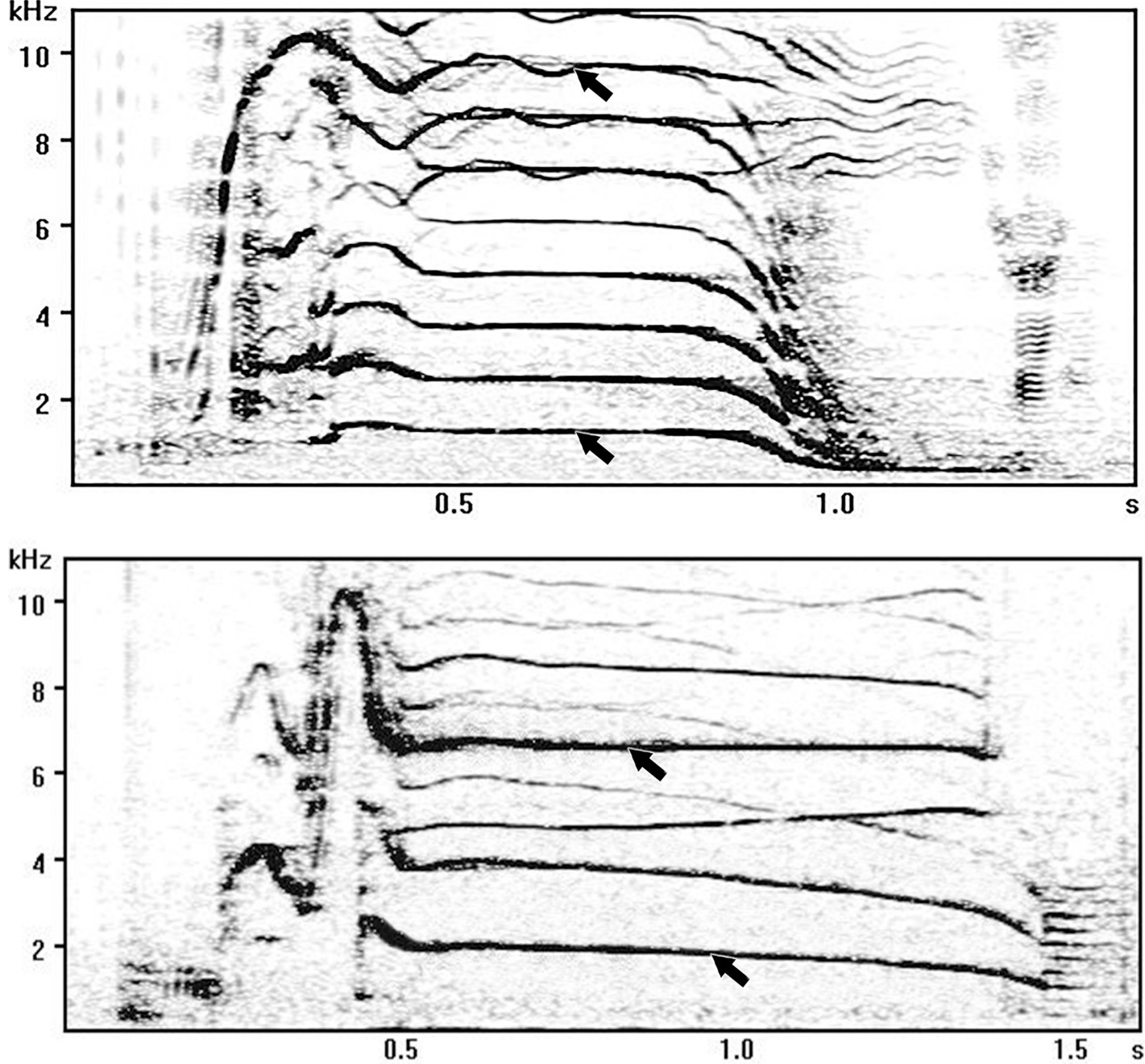

**Fig 1.** Locations of the amplitude measurements taken from the lower- and higher-frequency components of K5 (top) and K7 (bottom) call types.

adjust for the hearing threshold differences between the frequencies of the lower- and higher-frequency components. The audiograms of individual killer whales differed from each other, so I used the 'model' audiogram (Fig 3 in [39]) for the calculations. From the audiogram, I estimated the threshold at the mean frequency of each component of each call type. For K5 type, the mean frequency of the lower-frequency component was 1.1 kHz and of the higher-frequency component– 9.7 kHz, which corresponded to the thresholds of 93 and 60 dB re 1 µPa, respectively. Therefore, I assumed that the difference in the sensitivity to the lower- and

higher-frequency components of K5 type should be about 33 dB. For K7 type, the mean frequency of the lower-frequency component was 1.9 kHz and of the higher-frequency component– 6.5 kHz, which corresponded to the thresholds of 83 and 66 dB re 1 µPa, respectively. Therefore, I assumed that the difference in the sensitivity to the lower- and higher-frequency components of K7 type should be about 17 dB.

## Results

### Similarity of contours of the lower and higher frequency components

The similarity patterns of the lower and higher frequency components of both K5 and K7 calls across the families were largely unrelated (Fig 2 and S1 Table). The distance matrices of the lower and higher-frequency components of K5 and K7 calls had very weak correlation which was significant only for K5 (Mantel test, K5: $r = 0.089$, $p = 0.01$; K7: $r = 0.046$, $p = 0.099$).

Both for K5 and K7 call types distances between lower- and higher-frequency components from the same families and the same pods were smaller on average than between different pods (Table 1, Fig 2). Distances between the lower-frequency component contours of each call type were higher than the distances between the high-frequency contours for the same family/pod category of the same type, except for K7 call contours of the same families, where the relationship was reversed (Table 1).

The scatterplot of K5 call type distances clearly demonstrates the discrepancy between the similarity patterns of the lower- and higher-frequency components: there were many pairs of calls with highly different lower-frequency components and relatively similar higher-frequency components, and also many pairs of calls with dissimilar higher-frequency components and relatively similar lower-frequency components, but few calls that were highly dissimilar both in low- and higher-frequency components. This pattern did not occur in K7 call type.

Both K5 and K7 calls were classified more accurately to their respective family when both lower- and higher-frequency components were considered. For K5 calls, the correct classification rate was 70% when only the lower-frequency component was involved, 56% with only the higher-frequency component, and 81% when both components were used. For K7 calls, the correct classification rate for the lower-frequency component was 66%, for the higher-frequency component 51%, and for both components together 68%.

### Detectability of the lower and higher frequency components over distance

The higher-frequency component was detectable in most calls: only 16% of K5 calls and 5% of K7 calls had no visible higher-frequency component on the sonograms (S2 Table). All of the calls without detectable high-frequency components were distant and the amplitude of the lower-frequency component was also low, confirming that the higher-frequency component was not detectable due to transmission loss, and not because it was absent in the original call. All good-quality calls of these types had pronounced higher-frequency component. The frequencies of the higher-frequency component absorb roughly 1 dB/km more than the frequencies of the lower-frequency component; due to this effect, at 10 km distance, the higher-frequency component loose approximately 10 dB more than the lower-frequency component [7].

The received amplitude of the higher-frequency component was substantially lower than that of the lower-frequency component in K5 call type, but not in K7 call type (Fig 3). In K5 call type, the mean amplitude of the higher-frequency component was 17.5 dB lower than the mean amplitude of the lower-frequency component. In K7 call type, the mean amplitude of

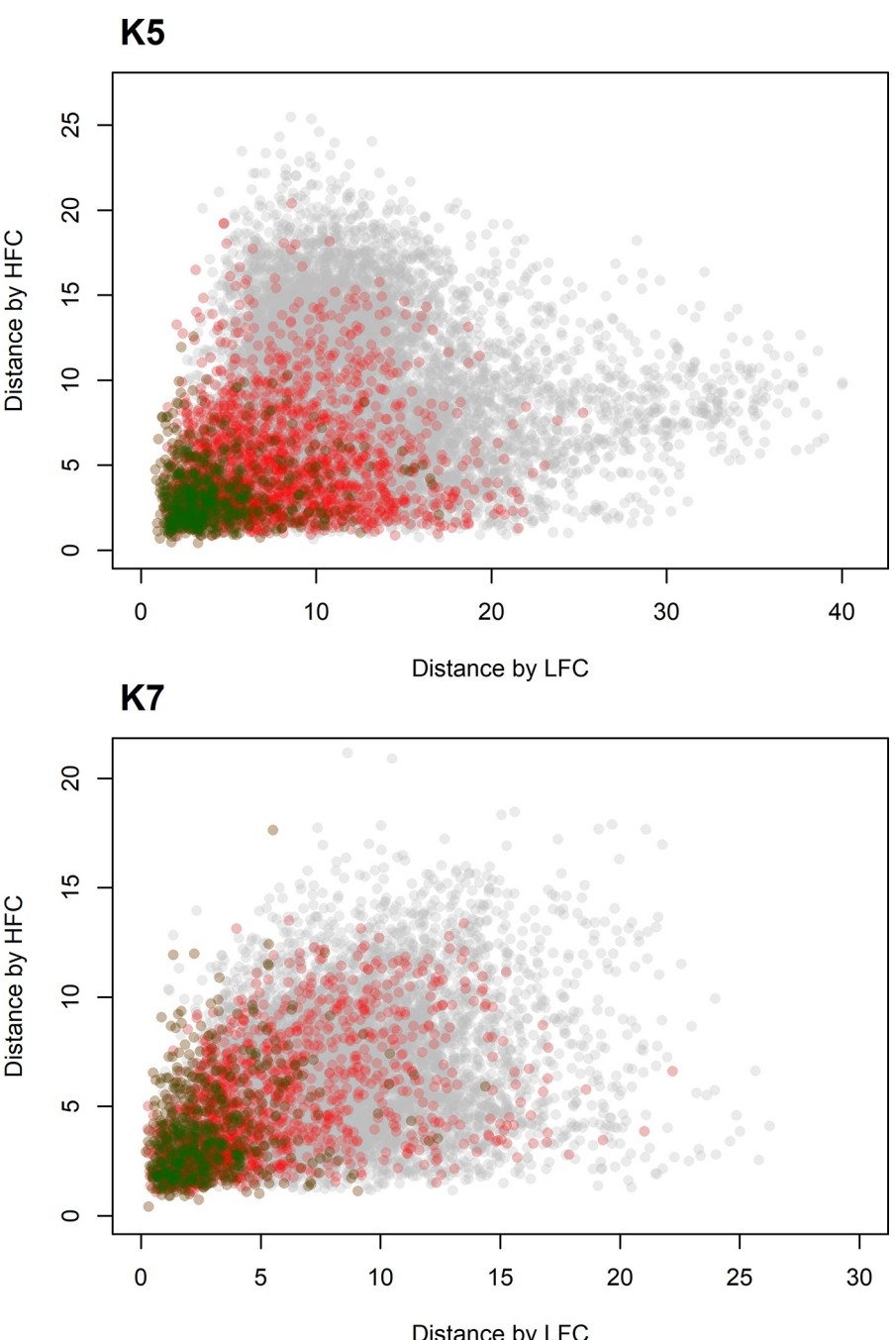

**Fig 2. Scatterplot of distances calculated through dynamic time warping for lower- and higher-frequency components of K5 (top) and K7 (bottom) call types.** Green—distances between the lower- and higher-frequency components from the same families; red–between different families from the same pods; grey–between different pods.

the higher-frequency component was only 0.1 dB lower than the mean amplitude of the lower-frequency component.

The higher-frequency component of K5 call type had higher frequency than that of K7 call type (mean±SD, K5: 9715±785 Hz; K7: 6505±305 Hz). Consequently, due to the higher hearing sensitivity of killer whales to the higher frequencies, more pronounced adjustment by the

**Table 1. Distances (mean ± SD) between pairs of calls calculated through dynamic time warping for lower- and higher-frequency components of K5 and K7 call types between the same families, between different families from the same pods, and between different pods.**

|  | K5 | | K7 | |
|---|---|---|---|---|
|  | **lower-frequency component** | **higher-frequency component** | **lower-frequency component** | **higher-frequency component** |
| Same families | 4.4 ± 2.8 | 3.4 ± 1.9 | 2.8 ± 1.9 | 3.8 ± 2.1 |
| Different families, same pods | 8.4 ± 4.2 | 5.4 ± 3.4 | 5.5 ± 3.7 | 5.1 ± 2.8 |
| Different pods | 12.3 ± 5.8 | 8.9 ± 4.9 | 8.2 ± 3.7 | 6.3 ± 2.9 |

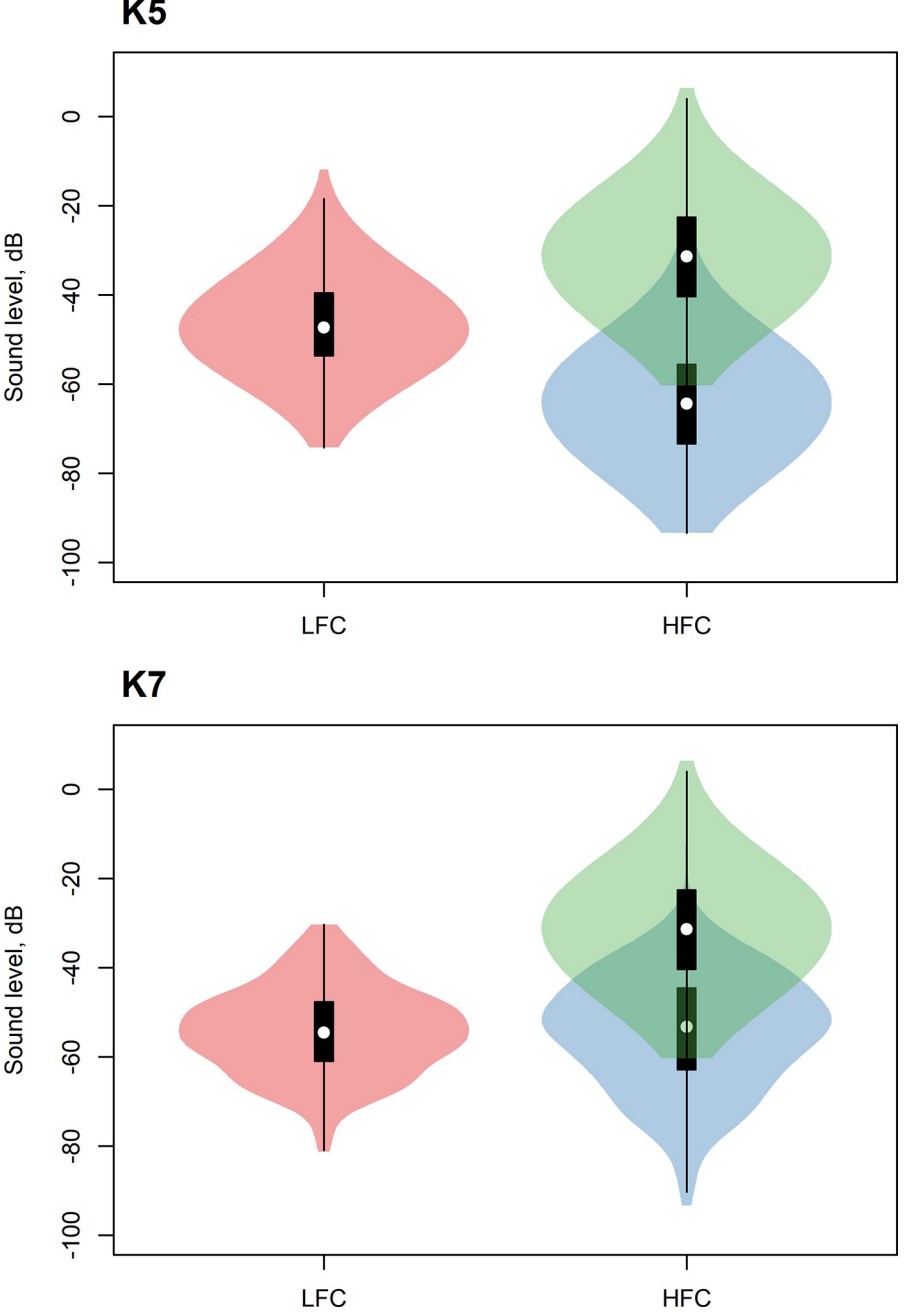

**Fig 3. Measured amplitude in the middle of the lower-frequency component (LFC, red) and in the middle of the higher-frequency component (HFC, blue) of K5 (top) and K7 (bottom) calls.** Green identify measurements of the higher-frequency component adjusted by killer whale audiogram.

hearing threshold was applied for the higher-frequency component of K5 call type (33 dB) than for that of K7 call type (17 dB). The perceived amplitude of the higher-frequency component of K5 call type after adjusting it by the hearing threshold was 15.5 dB higher, than the amplitude of the lower-frequency component. The perceived amplitude of the higher-frequency component of K7 call type after adjusting it by the hearing threshold was 16.9 dB higher, than the amplitude of the lower-frequency component.

## Discussion

In killer whales, the main function of biphonation has been hypothesized to be the better discrimination of a caller's orientation by listeners. Miller [40] has shown that when a calling killer whale is oriented towards a listener, the overlapping high frequency component bears substantially more energy, than when a caller is oriented away from a listener. It happens because high-frequency sounds are more directional in toothed whales, likely due to specific adaptations of their sound producing and enhancing structures. The difference in the relative energy of the lower and higher frequency components allows a listener to infer the direction of a caller's movement, which is crucial to the coordination of individuals on a distance. Underwater visibility is low compared to air; for example, in the North Pacific in summer it is hardly possible to see anything further than several meters. Biphonic sounds give the whales clues to identify the direction the other animals are moving. Moreover, the directionality of the higher-frequency component can provide additional benefits, such as allowing a caller to emit family/clan membership information towards specific whales or groups within an aggregation selectively.

However, if the orientation marking was the only function of the higher-frequency component, there would be no need to vary its shape across families: the same contour for all families would work equally well. Besides, the higher-frequency component is not the only element of acoustic repertoire that can indicate the orientation of the caller. Other sounds, such as echolocation clicks, burst-pulse sounds and even whistles also have directional properties [41].

In several terrestrial species, the overlapping higher-frequency component has been suggested to increase individual recognition. For example, it was shown that joining of two independent call components into a common vocalization may function to enhance individual recognition in the dhole [22]. In king and emperor penguins, biphonation enhances the accuracy of parent-chick and mate-mate recognition [21].

Additional hypotheses for the function of biphonation in other species have included: increasing unpredictability and providing an indication of physical condition [20]. However, these are likely not appropriate considerations as possible functions of biphonation in killer whales. These hypotheses suggest that the caller can choose when to include or not include a biphonic component in each call it makes. However, calls that comprise killer whale dialects are stereotyped, and if a call is biphonic, the presence of the higher-frequency component is obligatory, i.e. the same call type is not normally produced alternately with and without the higher-frequency component. Most likely, the higher-frequency component of killer whale biphonic calls functions as an alternative contour that bears information on the caller's identity (probably on the family level, in contrast to individual level in dholes and penguins). The presence of two independent components makes the recognition system twice more error-proof both in space and time. In space, if one of the components is not recognizable due to distance or masked by noise, the whales can still use the other component to identify the caller's family affiliation. In time, if in some families the contour shape of the higher- or lower-frequency component randomly converges through the process of cultural evolution [42], the other component still remains different between the families and allows listeners to discriminate between them.

The marine environment favors the acoustic channel for information transfer, because visibility underwater is very poor, while sound travels much further than in air. However, acoustic transmission has its limitations which are different for the low- and high-frequency sounds. Low-frequency sounds attenuate less and travel further than high-frequency sounds [43], therefore, in quiet conditions the lower-frequency component has better potential for long-range communication. On the other hand, underwater noise (both natural and man-made) is normally more pronounced in lower frequencies, often masking the lower-frequency component of killer whale calls.

Ship noise is a serious issue that can substantially reduce the detection distance of killer whale calls [44]. Killer whales can increase the amplitude of their calls in response to ship noise [45], but no studies have examined the relative noise immunity of the higher- and lower-frequency components of killer whale calls so far. Although at close ranges the energy of ship noise can extend above the higher-frequency component [46], normally it is mostly concentrated on low frequencies [47]. Therefore, it is likely that the higher-frequency component which lies above noise may function to facilitate pod and family recognition in noisy conditions. The directionality of the higher-frequency component can also help increase the signal-to-noise ratio in the direction of a receiver in noisy conditions. The amplitude comparisons in our study suggest that both higher- and lower-frequency components are equally important: the lower-frequency component is normally slightly louder, but this is compensated by the better hearing sensitivity of killer whales to higher frequencies [38, 39]. The higher-frequency component has higher sensation level, i.e. the whales normally hear the higher-frequency component slightly better than the lower-frequency component, which highlights its potential importance for their underwater communication.

Most biphonic calls of killer whales have heterodyne frequencies below and above the higher-frequency component (Fig 4). Heterodyne frequencies arise from the interaction of the two components, when the lower-frequency component is amplitude modulating the higher-frequency component [8, 12, 48]. This leads to the appearance of sidebands below and above the higher-frequency component that are equal to the difference or sum of the fundamental frequency of the higher-frequency component and the fundamental frequency and harmonics of the lower-frequency component [48]. Therefore, from the shape of the higher-frequency component and the heterodyne frequencies it is possible to infer the shape of the lower-frequency component when it is masked by low-frequency noise. It is unknown whether killer whales employ this capacity of biphonic calls, but it emphasizes the potential of the independently modulated higher-frequency component for underwater acoustic communication.

When both call components are detectable, biphonation can still be useful to duplicate the recognition potential of stereotyped calls, because the shape of call contours can randomly converge in unrelated social units [42]. Killer whale calls are not inherited genetically, but are rather learned from mother and other maternal relatives. Complex repertoires of stereotyped calls–vocal dialects–represent a form of animal culture [3]. Killer whale dialects slowly change in time through learning errors and innovations [2, 25, 49–51]; this process of cultural change is called cultural evolution [52, 53]. Since the variability of sound contours is limited due to the natural physical constrains, call contours of unrelated killer whale social units can sometimes become more similar due to the random convergence [42]. This would impede the discrimination between these two pods on a distance, when some call features are masked. The presence of an independently modulated and separately evolving call component would enhance recognition, because the probability that both components would converge in two pods is very low. Indeed, our results demonstrate that call discrimination by dynamic time warping was better when both components were used, compared to each component separately.

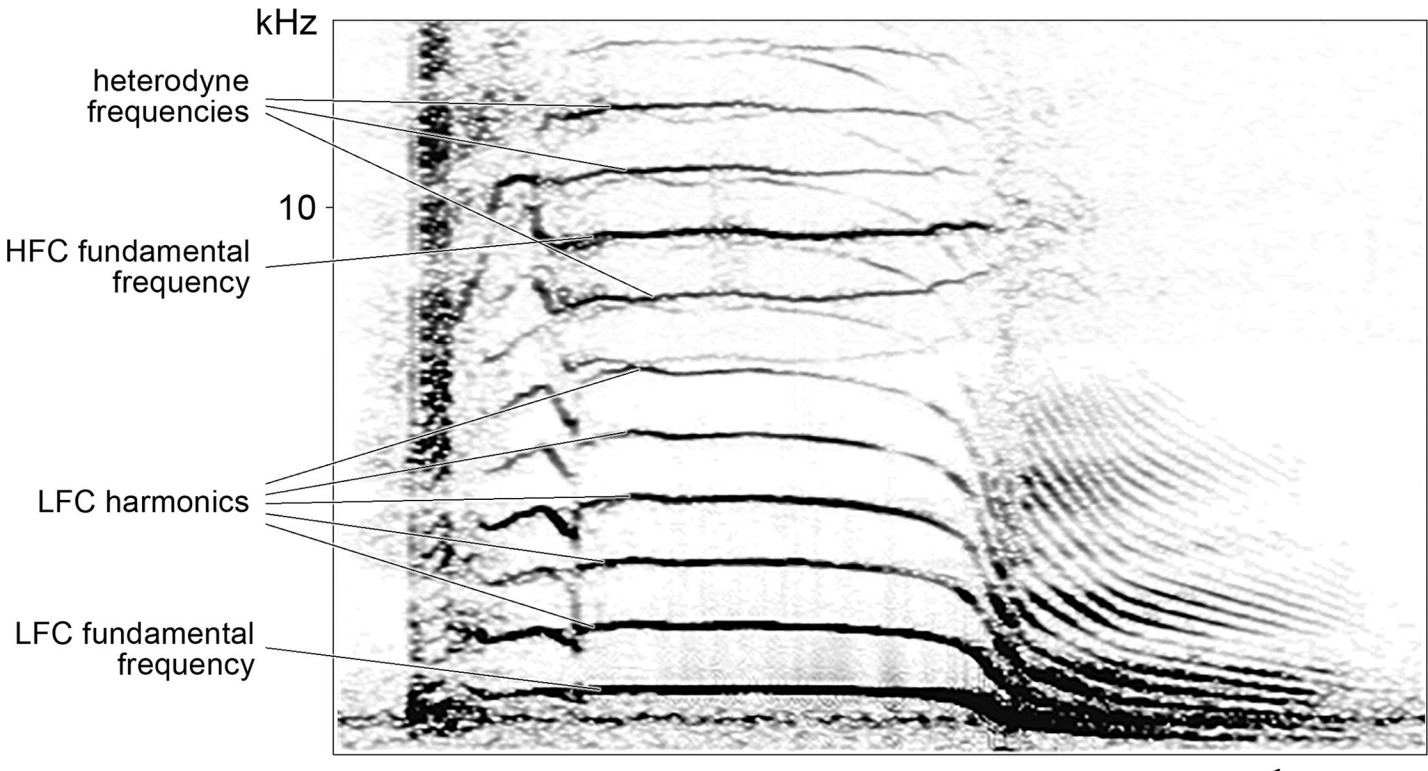

**Fig 4. Biphonic K5 call type showing heterodyne frequencies above and below the higher-frequency component.** Heterodyne frequencies arise from the interaction between the lower- and higher-frequency components; the frequency of the first heterodyne above the higher-frequency component is equal to the frequency of the higher-frequency component plus the frequency of the lower-frequency component, the frequency of the second heterodyne above the higher-frequency component is equal to the frequency of the higher-frequency component plus the frequency of the first harmonic of the lower-frequency component, and so on; the same principle applies to the heterodynes below the higher-frequency component, but with minus instead of plus.

Comparing the similarity patterns, I have found that they were different for the lower- and higher-frequency components. Many calls from different families had highly different higher-frequency component and relatively similar lower-frequency component and vice versa, but there were relatively few calls that were dissimilar in both components. This can indicate that dissimilarity in killer whale calls is costly and/or difficult to achieve because of the structural constrains [42], and for this reason it rarely occurs in both components.

Two examined call types had different patterns of similarity of the lower- and higher-frequency components: in K5 call type, adding the higher-frequency component to the contour substantially increased the rate of correct classification to family, while in K7 call type it increased the correct classification rate only slightly, suggesting that the higher-frequency component of K7 call type differs little across families. Different patterns of similarity of the lower- and higher-frequency components in different call types have been demonstrated previously for the eastern North Pacific killer whales: in some call types, adding the higher-frequency component increased the discrimination error, and in other call types, the effect was the opposite [54]. It is possible that the higher-frequency component of K7 serves the function of clan rather than family recognition. In Kamchatka, biphonic calls with higher-frequency component on 6–7 kHz, like K7, are found only in Avacha clan [34], which makes this clan easily acoustically recognizable on a distance. Yurk [55] also found significant differences across killer whale clans of Alaska and British Columbia in frequencies of the higher-frequency component, but not the lower-frequency component. This suggests that in some populations

the two components may serve for recognition on different levels–the lower-frequency component on the family level and the higher-frequency component on the clan level. This can happen if the components evolve with different speed. Variation in the rates of cultural evolution of different elements of killer whale vocal repertoire has been reported previously: Deecke et al. [25] showed that different call types can change with different speed, and Filatova et al. [5] found that different syllables within calls diverge with different speed, including syllables of both the lower- and higher-frequency components. Yurk [55] performed McDonald-Kreitman test for positive selection and found different selective pressures for the lower- and higher-frequency components. Overall, these findings suggest that independent cultural change of both components of biphonic calls may facilitate acoustic recognition of different levels of social structure (clans, pods or families) in killer whales. However, despite all of this evidence is consistent with independent evolution of the two components, it does not demonstrate it directly and rigorously. In future, analysis of the component variation over time is necessary to exclude the alternative drivers of the observed patterns.

In conclusion, I suggest that a likely function of the higher-frequency component is to duplicate and/or complement the social identity marking when the lower-frequency component is masked by noise or accidentally appears similar in unrelated social units. The combination of both components provides a redundancy of information that is beneficial for these animals to maintain contact over distance in the noisy underwater environment. The cultural evolution of two components of biphonic calls is happening independently of each other and can occur with different speed, which ensures the lack of correlation in their similarity patterns, providing the whales with an error-proof back-up mechanism to recognize the social affiliation of their conspecifics within distance of acoustic contact.

## Supporting information

**S1 Table. Similarity matrices for the lower- and higher-frequency components of K5 and K7 calls.**
(XLSX)

**S2 Table. Measurements of the amplitude of the lower- and higher-frequency components of K5 and K7 calls.** The table also shows the time stamp of the measurement within a file, and the frequency at which the amplitude was measured.
(XLSX)

## Acknowledgments

I am grateful to all members of FEROP expeditions, especially to Mikhail Guzeev and Anastasya Danishevskaya who were responsible for sound recordings, and to Tatiana Ivkovich for photo-identification of individual killer whales.

## Author Contributions

**Conceptualization:** Olga A. Filatova.

**Formal analysis:** Olga A. Filatova.

**Investigation:** Olga A. Filatova.

**Methodology:** Olga A. Filatova.

**Validation:** Olga A. Filatova.

**Visualization:** Olga A. Filatova.

**Writing – original draft:** Olga A. Filatova.

**Writing – review & editing:** Olga A. Filatova.

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
