## [Decision Letter · Decision Letter 0]

22 Apr 2020

PONE-D-20-05337

Independent cultural change of higher-frequency component can facilitate call recognition in killer whales

PLOS ONE

Dear Dr Filatova,

Thank you for submitting your manuscript to PLOS ONE. After careful consideration, we feel that it has merit but does not fully meet PLOS ONE’s publication criteria as it currently stands. Therefore, we invite you to submit a revised version of the manuscript that addresses the points raised during the review process.

We would appreciate receiving your revised manuscript by Jun 06 2020 11:59PM. To enhance the reproducibility of your results, we recommend that if applicable you deposit your laboratory protocols in protocols.io, where a protocol can be assigned its own identifier (DOI) such that it can be cited independently in the future. For instructions see: http://journals.plos.org/plosone/s/submission-guidelines#loc-laboratory-protocols

We look forward to receiving your revised manuscript.

Kind regards,

William David Halliday, Ph.D.

Academic Editor

PLOS ONE

Journal Requirements:

Additional Editor Comments (if provided):

Four reviewers have assessed this manuscript, and all agree that it has merit, but requires significant revision before it can be published. I agree with this assessment. Please address all comments from each reviewer when revising your manuscript. I look forward to seeing the revised manuscript. Reviewer 3 uploaded their comments in a file - please remember to download this file and address those comments.

Reviewers' comments:

Reviewer's Responses to Questions

**Comments to the Author**

1. Is the manuscript technically sound, and do the data support the conclusions?

Reviewer #1: Partly

Reviewer #2: Partly

Reviewer #3: Yes

Reviewer #4: Partly

2. Has the statistical analysis been performed appropriately and rigorously? 

Reviewer #1: Yes

Reviewer #2: Yes

Reviewer #3: Yes

Reviewer #4: Yes

3. Have the authors made all data underlying the findings in their manuscript fully available?

Reviewer #1: Yes

Reviewer #2: Yes

Reviewer #3: Yes

Reviewer #4: Yes

4. Is the manuscript presented in an intelligible fashion and written in standard English?

Reviewer #1: Yes

Reviewer #2: Yes

Reviewer #3: Yes

Reviewer #4: Yes

5. Review Comments to the Author

Reviewer #1: The manuscript describes an important bio-acoustical subject, the use of bi-phony (two-voiced calls) by killer whales and suggest possible functions.

The topic has been subject of interest not just within the killer whale or cetacean research community but is of interest to the much broader research community interested in the evolution of communication signals and is specifically addressing adaptive function of signal structure used not just by killer whales.

The manuscript is well written and understandable for an audience with some background in bioacoustics and also provides explanations of some of the more complex bio-acoustic phenomena such as heterodyne frequencies in two component pulsed vocalizations. As such the manuscript provides understandable information for a wider audience of PlosOne.

The authors provide a number of intriguing suggestions in support of the main topic of the manuscript as stated in the title “Independent cultural change of higher-frequency component can facilitate call

recognition in killer whales” and support some of their main conclusions, such as the complementary function of both the lower and higher frequency call components in family recognition. The authors provide support for that conclusion in their investigation of family classification accuracy of two calls of resident type killer whales in the Russian Far East by considering classification means of the two call components separately. The argument that both components can be used independently to identify family membership of the caller is well laid out. The author also provide good support from their data that calls that have both components visible in the spectrograms have higher classification means than those where only have on of the two components is visible thereby validating one of the main conclusions of the study that calls with more than one components, especially those with heterodyne frequencies, have a higher ability to be used as family group identifiers in a variety of acoustic environments.

The authors argument, however, that the two-component structure of calls provides better propagation capabilities is only weakly supported by the data provided. The reasons for my assessment is based on the physics of underwater acoustics that make it difficult to interpret the data analysis in the way the authors did.

The calls collected for the analysis likely were collected on a number of occasions taking place at different locations and at different times of the year with some consistency in the latter due to field work seasonality. Spectral sound propagation loss varies tremendously between locations, i.e. different frequencies attenuate very differently at different locations due to things like bathymetry, sediment structure and water properties, the latter varies not only spatially but also temporally (e.g. vertical sound speed profiles can vary within hours at some locations depending on tidal, wind, and water mixing conditions). We can assume that the calls were not produced all at the same depths although based on tagging studies resident killer whales may produce most of their calls in the upper water column (< 20 m water depths). This upper layer of water is often varying in sound propagation conditions mostly due to temperature fluctuations.

All of the above introduces variation in the propagation of the different components that cannot be confounded by using received levels at one location with on hydrophone. Although the authors tried to minimize this variation by only looking at the sound pressure differential between the frequency components, the reality is that this variation can be have a much higher magnitude between locations and seasons than is considered by the authors. What this means that the same call components produced by the same animal may propagate very differently relative to each other in different locations and at different times in the same location.

Furthermore, the authors weigh the received levels of these components by assigning different components different sensitivities based on the hearing curve of the animals. While frequency based hearing sensitivity definitely plays a big role in the detection of the call components it is unlikely that killer whales can detect sound pressure differences linearly but their hearing is dependent on the auditory filter bandwidth that applies to the specific auditory system. Usually, that would be an octave fractal, such 1/3, 1/5, 1/6 or even 1/12 octaves auditory filter bandwidth over which the animal integrate sound pressure. So, while it is definitely true that higher frequency components are detected according to the higher hearing sensitivity of killer whales for those frequencies, we cannot assign a numerical value ( the authors chose 33 dB) as a weighting pressure when comparing perception of low and high frequency components. Since fractal filters are proportional filters, they become wider with increasing frequencies, which means the animals may be able to detect smaller pressure differences in lower frequencies while pressure differences for higher frequencies need to be more pronounced for the animal to perceive.

So, I don’t think the different propagation leads to the described effects in detection differential described by the authors. So, this section on propagation difference would need be revised to include uncertainty in the weighting assumption and the conclusions based on that section should also be revised.

Reviewer #2: Comments PONE-D-20-05337

General

Multiple publications have hypothesized the functions of biphonic calls in resident killer whales through contextual use of these calls and call features using source levels: group identity, contact over large distances, and inferring direction of the caller. This study uses recordings from Russian resident killer whales to test two of these hypotheses: group identity and contact over longer distances.

Throughout the manuscript the author refers to ‘killer whales’ when what is more correct is resident killer whales or fish eating killer whales

The author uses assignment to family group based on the low frequency component (LFC), high frequency component (HFC), and the LFC and HFC together to test the hypothesis that biphonic calls provide information on group identity. Wouldn’t one expect more information usually provide better classification? The LFC appears to account for the majority of correct classification when the two are combined, and the addition of the HFC only minimally improves classification success. Wouldn’t comparing the success rate of classification of biphonic and monophonic calls be more appropriate to test this hypothesis?

The author uses a large set of calls to compare the received levels of the LFC and HFC to test long range detectability each. As mentioned in the discussion, there are many things that can impact the detectability of low and high frequency sounds: direction of the caller, distance to the caller, spreading loss, and background noise. These can impact the LFC and HFC in different directions, therefore having an impact on the relative amplitude of the LFC and HFC. It is unclear how the authors account for this. The large sample size, and recording calls across a variety of scenarios, may be adequate, but the author need to address this.

From the methods presented it is unclear how the adjustment for the hearing threshold was done. My understanding is that the relative amplitude differences of the LFC and HFC were adjusted based at the hearing sensitivity at those frequencies. But this would require absolute received levels of the calls, which we do not have for this study (see above). This section of the methods needs more details.

Specific:

Introduction

Par1,line 2: pod and or community?. Maybe say every group of killer whales or say “In resident killer whales, each killer whale pod….”

Par2,line 3: evolve should be evolves

Par2, line 6: instead of the “- “ start a new sentence

Par3, line 6: ‘Besides’ is awkward wording here. ‘Additionally’ or ‘furthermore’

Par5, line 1-2: again this is the case only for resident killer whales. Others, like Bigg’s and offshores disperse.

Materials and Methods

Similarity of contours of the lower and higher frequency components section:

You mention the number of families, but how many different pods does this represent?

Were the 10 calls quality graded as in Deecke and Janik? Were the calls of highest quality? Adequate signal to noise ratio?

Paragraph 3- is this final number what is referred to as distances in the results? This should be clarified.

Detectability of the lower- and higher-frequency components over distance section

Wouldn’t the direction of the caller impact the amplitude of the HFC of calls more than the LFC?

Recording quality and signal to noise ration can impact the ability to make reliable measurements. If the dataset does not have high levels of background noise that would impact these measurements, some clarification/quality grading/ analysis of impact should be done.

Calls with no detectable HFC… it should be clarified that these weren’t used in the analysis

Results

Detectability of the lower and higher frequency components over distance section:

The mean difference in amplitude between the HFC and LFC is reported but the range or SD should also be reported.

Reviewer #3: I have uploaded my review comments as a separate document because it exceeds 20000 characters (all comments are geared towards providing constructive feedback to help improve the manuscript and arguments). Overall, the manuscript is well written, but would benefit from copy editing to improve syntax and grammar. Please refer to my attachment for further comments and details.

Reviewer #4: Overall this is a valuable study that advances our understanding of this acoustic phenomenon in an important way. The conclusion that “the main function of the higher-frequency component is to duplicate and/or complement the social identity marking when the lower-frequency component is masked by noise or accidentally appears similar in unrelated social units.” Is well supported by the analyses and so I recommend the paper be accepted subject to some changes needed in interpretation and presentation.

My biggest concern is that the Title and second primary conclusion is not adequately justified as ‘independent cultural change’ has not been shown- that can only be done using a temporal study showing how the features have changed over time. The pattern found could just be due to natural variability within otherwise fixed contours – or contours that slowly evolve but not independently. New title could be: Independent acoustic variation of higher-frequency components can facilitate call recognition in killer whales

The manuscript is important without this unsupported claim. You should propose specific future work by which the question of whether or not these call types truly evolve independently is tested more directly and robustly.

A second overall concern is that the write-up doesn’t cite all of the most relevant work in the field. It is important to cite and discuss those published studies that most closely relate to your current work.

Minor suggestions:

Ln 39 - suggest you delete ‘excellent’. The cited study shows rather poor copies of sounds made by one captive killer whale.

Ln 40 add ‘call repertoire’ before dialect to be more precise.

Ln 46-‘evolves’

Ln 49 – no ‘the’ before ‘social’

Ln 55- A paper by Deecke The structure of stereotyped calls reflects kinship and social affiliation in resident killer whales (Orcinus orca) - seems important to cite here.

Ln 59 or 62. A paper by Miller Caller sex and orientation influence spectral characteristics of “two-voice” stereotyped calls produced by free-ranging killer whales -seems important to cite here.

Ln 149-it is strange to read ‘we’ for a single authored paper.

Ln 158 and Discussion: One highly relevant paper that isn’t cited here looked at error rates of classification for LFC versus HFC: “The influence of social affiliation on individual vocal signatures of northern resident killer whales (Orcinus orca)” It appears that very similar methods and results were found, though on a slightly different social scale.

Methods: consider the possibility that the ‘same family’ results may have been influenced by some of the calls being produced by the same individual.

Ln 225 and 226 – add ‘pairs of’ before ‘calls’

Table1and Figure 2 – what are the units here? This value seems to come from a black box, so make an effort to convince the reader that it is a valid indicator of the similarity.

Table 1- this is a mean of the distances averaged across a lot of pairs of calls, right? If so, state that this is a mean value, and add the standard deviation and sample size to the table.

Table 1 title is confusing- State that is the distance between pairs of calls.

Figure 2-consider to use smaller symbols and open face symbols to better show the data underneath other data points.

Figure 2 – add an x-axis label. Correct top panel y-axis label.

Ln 240-246: Did you ever see cases when only the higher component was visible?

Discussion:

The difference between a signal and the audiogram sensitivity at that frequency is commonly known as the ‘sensation level’. This is a dictionary definition, and could be useful for your paper.

Ln 309 – how much more quickly does the higher frequency component attenuate than the lower frequency component-in dB/km? Is that difference enough to overcome the higher sensation level of the higher component with distance? (personally I think the difference is very small, but it is worth to show you looked at this).

LN 331-333: Again here it’s relevant to cite Miller et al Caller sex and orientation influence spectral characteristics of “two-voice” stereotyped calls produced by free-ranging killer whales -- as it supports the presence of heterodyne frequencies in the two component calls from a totally different population of killer whales.

Ln 336-338. I’d suggest caution here regarding your point on evolution of call types. It is clear that we don’t know the actual mechanism by which these contours are produced, which limits our ability to be certain that each component can be modified independent of the other. There may be aspects of sound production that fundamentally limit the flexibility of one component to change without affecting the other.

Ln 399- all of this evidence is consistent with independent evolution of the two components, but doesn’t demonstrate it directly and rigorously. Alternative drivers of the patterns you found remain possible. Stronger to here to indicate future temporal analyses that would be able to tackle the question more directly.

Acknowledgements reads odd with ‘We’ if this is a single author work.

Overall, very good work.

6. PLOS authors have the option to publish the peer review history of their article (what does this mean?). If published, this will include your full peer review and any attached files.

Reviewer #1: Yes: Harald Yurk

Reviewer #2: No

Reviewer #3: No

Reviewer #4: No

---

## [Author Response · Author response to Decision Letter 0]

28 May 2020

Dear Reviewers,

thank you very much for your time and the thorough assessment of my manuscript. I have substantially revised it following your corrections and suggestions. I have changed the title and added the clarification about the ecotype of the study population. I have added the relevant references and added more details to the methods regarding the call recording, selection and analysis. I have expanded the discussion as suggested by the reviewers. Overall, I am very grateful to all of you for your detailed corrections, which I believe have substantially improved the manuscript.

Reviewer #1: The manuscript describes an important bio-acoustical subject, the use of bi-phony (two-voiced calls) by killer whales and suggest possible functions.

The topic has been subject of interest not just within the killer whale or cetacean research community but is of interest to the much broader research community interested in the evolution of communication signals and is specifically addressing adaptive function of signal structure used not just by killer whales.

The manuscript is well written and understandable for an audience with some background in bioacoustics and also provides explanations of some of the more complex bio-acoustic phenomena such as heterodyne frequencies in two component pulsed vocalizations. As such the manuscript provides understandable information for a wider audience of PlosOne.

The authors provide a number of intriguing suggestions in support of the main topic of the manuscript as stated in the title “Independent cultural change of higher-frequency component can facilitate call recognition in killer whales” and support some of their main conclusions, such as the complementary function of both the lower and higher frequency call components in family recognition. The authors provide support for that conclusion in their investigation of family classification accuracy of two calls of resident type killer whales in the Russian Far East by considering classification means of the two call components separately. The argument that both components can be used independently to identify family membership of the caller is well laid out. The author also provide good support from their data that calls that have both components visible in the spectrograms have higher classification means than those where only have on of the two components is visible thereby validating one of the main conclusions of the study that calls with more than one components, especially those with heterodyne frequencies, have a higher ability to be used as family group identifiers in a variety of acoustic environments.

The authors argument, however, that the two-component structure of calls provides better propagation capabilities is only weakly supported by the data provided. The reasons for my assessment is based on the physics of underwater acoustics that make it difficult to interpret the data analysis in the way the authors did.

The calls collected for the analysis likely were collected on a number of occasions taking place at different locations and at different times of the year with some consistency in the latter due to field work seasonality. Spectral sound propagation loss varies tremendously between locations, i.e. different frequencies attenuate very differently at different locations due to things like bathymetry, sediment structure and water properties, the latter varies not only spatially but also temporally (e.g. vertical sound speed profiles can vary within hours at some locations depending on tidal, wind, and water mixing conditions). We can assume that the calls were not produced all at the same depths although based on tagging studies resident killer whales may produce most of their calls in the upper water column (< 20 m water depths). This upper layer of water is often varying in sound propagation conditions mostly due to temperature fluctuations.

All of the above introduces variation in the propagation of the different components that cannot be confounded by using received levels at one location with on hydrophone. Although the authors tried to minimize this variation by only looking at the sound pressure differential between the frequency components, the reality is that this variation can be have a much higher magnitude between locations and seasons than is considered by the authors. What this means that the same call components produced by the same animal may propagate very differently relative to each other in different locations and at different times in the same location.

Response: I agree that our recordings varied greatly in sound propagation properties, but I think that given the large sample size and time span, this is actually a good thing: this means that we likely captured most of the variation that killer whales themselves experience while listening to conspecific sounds, at least in the coastal waters in summer. I have included this explanation into the methods.

Furthermore, the authors weigh the received levels of these components by assigning different components different sensitivities based on the hearing curve of the animals. While frequency based hearing sensitivity definitely plays a big role in the detection of the call components it is unlikely that killer whales can detect sound pressure differences linearly but their hearing is dependent on the auditory filter bandwidth that applies to the specific auditory system. Usually, that would be an octave fractal, such 1/3, 1/5, 1/6 or even 1/12 octaves auditory filter bandwidth over which the animal integrate sound pressure. So, while it is definitely true that higher frequency components are detected according to the higher hearing sensitivity of killer whales for those frequencies, we cannot assign a numerical value ( the authors chose 33 dB) as a weighting pressure when comparing perception of low and high frequency components. Since fractal filters are proportional filters, they become wider with increasing frequencies, which means the animals may be able to detect smaller pressure differences in lower frequencies while pressure differences for higher frequencies need to be more pronounced for the animal to perceive.

Response: This comment would be relevant if I was assessing the ability of whales to detect pressure differences across some frequency range. However, this was not the case – I was looking for the hearing sensitivity of killer whales at some particular frequencies separately. I looked at the audiogram (model audiogram at Fig. 3 in Branstetter et al. 2017) and saw that the threshold at the frequency of the lower component (1.1 kHz) was about 93 dB re 1 µPa, and at the frequency of the higher component (9.7 kHz) it was about 60 dB re 1 µPa. Therefore, the difference in the sensitivity to the lower and higher components should be about 33 dB. I have added more details of this process to the methods.

So, I don’t think the different propagation leads to the described effects in detection differential described by the authors. So, this section on propagation difference would need be revised to include uncertainty in the weighting assumption and the conclusions based on that section should also be revised.

Reviewer #2: Comments PONE-D-20-05337

General

Multiple publications have hypothesized the functions of biphonic calls in resident killer whales through contextual use of these calls and call features using source levels: group identity, contact over large distances, and inferring direction of the caller. This study uses recordings from Russian resident killer whales to test two of these hypotheses: group identity and contact over longer distances.

Throughout the manuscript the author refers to ‘killer whales’ when what is more correct is resident killer whales or fish eating killer whales

Response: I have changed “killer whales” to “resident-type (R-type) killer whales” when appropriate (we try not to use terms ‘resident’ and ‘transient’ when referring to Russian killer whales because it is confusing for our officials, so we adopted the terms ‘R-type’ and ‘T-type’ instead).

The author uses assignment to family group based on the low frequency component (LFC), high frequency component (HFC), and the LFC and HFC together to test the hypothesis that biphonic calls provide information on group identity. Wouldn’t one expect more information usually provide better classification? 

Response: Not necessarily. If all HFCs were identical, adding them to the analysis would not improve the classification success. If they had different similarity pattern unrelated to family affiliation (for example if some HFCs in family 1 were more similar to family 2, and other to family 3), adding them would in fact decrease the correct classification rate.

The LFC appears to account for the majority of correct classification when the two are combined, and the addition of the HFC only minimally improves classification success. Wouldn’t comparing the success rate of classification of biphonic and monophonic calls be more appropriate to test this hypothesis?

Response: No, it wouldn’t. The problem is that monophonic and biphonic calls in killer whales differ not just by the presence of the higher-frequency component; they have different source levels, different diversity, and their usage varies depending on the behavioral context, suggesting different functions. Therefore, we can expect that their group-specificity can be also different, and therefore we cannot extrapolate results from monophonic calls to the lower-frequency component of biphonic calls. 

The author uses a large set of calls to compare the received levels of the LFC and HFC to test long range detectability each. As mentioned in the discussion, there are many things that can impact the detectability of low and high frequency sounds: direction of the caller, distance to the caller, spreading loss, and background noise. These can impact the LFC and HFC in different directions, therefore having an impact on the relative amplitude of the LFC and HFC. It is unclear how the authors account for this. The large sample size, and recording calls across a variety of scenarios, may be adequate, but the author need to address this.

Response: I have added a paragraph on this issue to the Methods: “The calls were collected for the analysis at different locations within Avacha Gulf. Spectral sound propagation loss can vary between locations due to differences in bathymetry, sediment structure and water properties; some variation is also introduced by differences in depths at which killer whales produce their calls. To address this, we measured a large amount of calls produced in various conditions and in different situations to cover as much natural variation as possible. As killer whales also experience all these propagation effects when listening to the conspecific calls, we assume that our sample set is an adequate representative of what killer whale normally hear in the wild.”

From the methods presented it is unclear how the adjustment for the hearing threshold was done. My understanding is that the relative amplitude differences of the LFC and HFC were adjusted based at the hearing sensitivity at those frequencies. But this would require absolute received levels of the calls, which we do not have for this study (see above). This section of the methods needs more details.

Response: I have added more details on this to the methods. In short, I estimated the difference in thresholds at the frequency of the lower and higher components, and then added this difference to the received level of the higher component of each call.

Specific:

Introduction

Par1,line 2: pod and or community?. Maybe say every group of killer whales or say “In resident killer whales, each killer whale pod….”

Response: I changed it to “Each resident-type (R-type) killer whale pod”.

Par2,line 3: evolve should be evolves

Response: Changed, thank you!

Par2, line 6: instead of the “- “ start a new sentence

Response: Done.

Par3, line 6: ‘Besides’ is awkward wording here. ‘Additionally’ or ‘furthermore’

Response: Changed to ‘Additionally’.

Par5, line 1-2: again this is the case only for resident killer whales. Others, like Bigg’s and offshores disperse.

Response: I added ‘R-type’ here to clarify this.

Materials and Methods

Similarity of contours of the lower and higher frequency components section:

You mention the number of families, but how many different pods does this represent?

Response: I added this information to the text.

Were the 10 calls quality graded as in Deecke and Janik? Were the calls of highest quality? Adequate signal to noise ratio?

Response: The calls were not quality graded, but we used only the highest-quality calls with all syllables clearly visible. 

Paragraph 3- is this final number what is referred to as distances in the results? This should be clarified.

Response: Sorry, I should have clarified this: the distances are the opposite of the similarities, i.e. 100 minus the final number obtained through dynamic time warping. I have added this information to the text. 

Detectability of the lower- and higher-frequency components over distance section

Wouldn’t the direction of the caller impact the amplitude of the HFC of calls more than the LFC?

Response: It would, but I assume that given the large sample size, I will get the relatively uniform distribution of caller’s orientation towards the hydrophone, which will be a good approximation of what killer whales normally hear. 

Recording quality and signal to noise ration can impact the ability to make reliable measurements. If the dataset does not have high levels of background noise that would impact these measurements, some clarification/quality grading/ analysis of impact should be done.

Response: Actually, it doesn’t. I took very simple measurements – amplitude in the middle of the lower and higher-frequency components. They are pretty obvious and easily obtained even from calls with low SNR. However, most calls were of reasonable quality, because we normally do not do recordings in presence of loud noise. 

Calls with no detectable HFC… it should be clarified that these weren’t used in the analysis

Response: I added this clarification.

Results

Detectability of the lower and higher frequency components over distance section:

The mean difference in amplitude between the HFC and LFC is reported but the range or SD should also be reported.

Response: I have added SDs to the table.

Reviewer #3: I have uploaded my review comments as a separate document because it exceeds 20000 characters (all comments are geared towards providing constructive feedback to help improve the manuscript and arguments). Overall, the manuscript is well written, but would benefit from copy editing to improve syntax and grammar. Please refer to my attachment for further comments and details.

Overall Review Summary:

Overall, the study meets PLOSone’s Publication Requirements. I believe the study provides important insights into the potential functionality of biphonation in resident killer whale vocalizations and will be an important addition to the available literature on the subject – I recommend it be published, after the comments in this review are taken into account. The study is well executed. The field efforts to collect this acoustic data for the study for so many different family groupings is impressive and worthy of recognition. The author does an adequate job of citing and reviewing relevant literature on the subject from multiple taxa throughout the manuscript. The study was performed using established methods that have been published previously, with some modification. I would have liked to see this study include more than one population of resident killer whale to increase scope, such as also including recordings from the Northern Resident population, but this is not essential and the study still contributes in a meaningful way.

Overall, the manuscript is well written, but would benefit from copy-editing to address some issues in grammar and syntax. There are some sections that require re-wording for clarification and interpretability (see line-by-line comments below). 

I may use the abbreviations ‘HFC’ and ‘LFC’ in some portions of my review, these just refer to ‘higher frequency component’ and ‘lower frequency component’, respectively.

General comment:

The author makes many references to ‘killer whales’, in general, throughout the manuscript, but the statements really only apply to a given ecotype (most often, ‘resident’ killer whales). And because the author only assesses vocalizations for resident-type killer whales in this study, the conclusions cannot be generalized over all killer whales/types due to the different acoustic behaviour, acoustic population sub-structuring, and vocalizations that other ecotypes exhibit. I suggest the author clarify their use of ‘killer whale’ in the paper to limit it to just the resident ecotype where appropriate.

Response: To clarify this, we have added ‘resident-type’ or ‘R-type’ to the text when appropriate. We are trying to avoid using the term ‘resident’ when referring to Russian residents because this provides too much confusion when discussing the ecotypes with our officials. Therefore, over the last 2-3 years we have been using mostly the terms ‘R-type’ and ‘T-type’ instead of ‘resident’ and ‘transient’ in our papers. 

The author also makes many references to ‘pod’ or ‘pods’ throughout the manuscript, but this can be misleading and ambiguous without providing a definition of the term to relate it to the previously published definitions of the term. See comments below for discussion on this. 

 Response: See the response about the ‘pod’ definition below.

Line-referenced Comments:

Title: I think the title can more accurately reflect the messages of the manuscript. First, it only speaks to the high frequency component, but the paper argues that the presence of both the high- and low-frequency component together provide killer whales with a tool that will ensure transmission of information in many more conditions/situations than either component individually. Second, the title only speaks to ‘call recognition’ but the paper’s main arguments revolve around the information that the calls carry – ie. Social group affiliation at various resolutions. Third, the title generalizes across killer whales, whereas I do not think these results can be interpolated beyond the ‘resident’ ecotype. I may recommend the following title (or something similar) to more accurately reflect the paper’s outcomes/arguments and reach: “Independent culturally-induced change of the higher- and lower-frequency components of biphonic calls can facilitate call recognition and social affiliation in resident killer whales”. This title more accurately represents the author’s conclusions in the discussion, and the fact that this really only relates to the resident killer whale ecotype (until similar research is conducted on the other ecotypes the use of the biphonation cannot be assumed to be the same without further study because different ecotypes seem to have different acoustic population sub-structuring, and thus the information communicated with the biphonation and the contexts it is used could be quite different).

Response: I changed the title to “Independent acoustic variation of the higher- and lower-frequency components of biphonic calls can facilitate call recognition and social affiliation in killer whales”. I removed the reference to cultural change from the title according to the comment of reviewer #4.

Line 12: the author’s use of the term ‘pod’ needs to be defined. See further comments on this topic below.

Response: See the response about the ‘pod’ definition below.

Lines 28 and 29: Change “…component allows to recognize the family identity…” to “…component allows the recognition of family identity of a caller…”.

 Response: Changed.

Line 40: The author uses the term ‘pod’ throughout the manuscript, however, the meaning of ‘pod’ is ambiguous. The author’s intended meaning of the term ‘pod’ needs to be defined. “Pod” has several meanings in the context of killer whales. “Pod” can be used in the traditional sense to describe a group of cetaceans (like a ‘herd’ of cattle, or ‘flock’ of birds). However, in killer whales, ‘pod’ was also defined by Bigg et al. 1990 as a functional social unit within the nested social structure of resident killer whales in British Columbia, as “A subpod made of groups who spend more than 50% of the time together“. Although this definition has been adopted by many researchers monitoring other killer whales around the world to describe social groupings, Ford and Ellis (2002, Reassessing the social organization of resident killer whales in British Columbia) determined that the ‘pods’ within Northern Resident Killer whales that were grouped base on the definition in Bigg 1990 no longer met this definition of ‘pod’ after ~30 years of monitoring the population (ie. a “pod” according to the social definition in Bigg et al 1990 is not a stable social grouping, and thus the use of the term ‘pod’ in this context should no longer be used). The author should define their intended meaning of ‘pod’ throughout the paper, or use a different descriptor. Or if the social groupings in their study population still adhere to Bigg’s definition of ‘pod’, then describe this (although, as mentioned, even though the population may still adhere to the definition, it likely won’t always, and a different term should be used). 

Response: Well, it is a good point. Actually, we had found that we cannot use Bigg’s 50% definition of pod for our population, because the associations between matrilineal units are more fluid: some units can spend 90% of time together during one summer and only 10% of time together next summer. Therefore, we rather adopted Ford’s dialect-based definition of pod as a set of matrilines that share the same dialect. I referred to this definition in the Methods: “Families that share the same vocal dialect are attributed to the same pod, and pods with similar dialects form clans.” To clarify the term earlier in the text, I added the dialect-based definition to the Introduction: “Each R-type killer whale pod shares the same dialect”

Line 52: “New pods form gradually through the split of an ancestral pod after a matriarch dies…”. First, the use of ‘pod’ needs to be defined or not used (use ‘group’ instead), as described in comment above. Second, this is not the only mechanism that drives the creation of new groups in resident killer whales. New groups also form through group-splitting that takes place while the maternal ancestor is still alive (intra-geneological splitting), as described in Stredulinsky 2016: Determinants of Group Splitting in a Threatened Population of Fish-eating Killer Whales. Group splitting is a more complex process than this sentence suggests, and as such, so is dialect evolution.

Response: I have deleted “after a matriarch dies” statement. Now it reads “New pods form gradually through the split of an ancestral pod, and the dialects of the splitting pods slowly diverge in time due to changes in call structure”. Of course it is a simplistic description, but it is accurate enough to give an idea of the process to a reader who is new to the field, and provides enough background to comprehend the further inferences in the text.

Line 73-75: Strongly suggest a change in wording. Natal philopatry is not present in all types of killer whales (therefore the author should only relate this sentence to resident killer whales, not killer whales in general). Also, I would argue that family life is not ‘critical’ for survival in killer whales. Family cohesion has some very notable benefits that make maintaining family cohesion more advantageous than not in many killer whale societies, but it is not critical. There are some killer whale societies where emigration out of one’s family is important (such as in the West Coast Transient “Bigg’s” killer whale population). The author should refrain from generalizing across killer whales generally, and try to stick within the study population (resident killer whales).

Response: I have added ‘R-type’ here to clarify that it applies only to resident-type killer whales, and replaced ‘critical’ with ‘important’.

Line 77: ‘Intentionally’ and ‘deliberately’ are synonyms.

 Response: I have deleted “intentionally”.

Line 113-114: Were the hydrophones calibrated across frequencies? What method did the author use to determine the reportedly flat response (manufacturers documentation/technical specifications of the equipment? End-to-end system calibration including recording equipment? Hydrophone-only calibration?). If the hydrophones were not calibrated, this needs to be stated, with an indication as to how the flat response was determined.

Response: For the Offshore Acoustic hydrophone, we rely on the manufacturer’s specifications, and the CetaPhone hydrophone was calibrated by us in combination with Zoom H4 flash-recorder. However, I checked specifically the recordings I used for this study and found that they were all made with the Offshore Acoustic hydrophone. I have added its manufactuter’s specifications to the Methods.

Line 108-135: Were any observations about group behavioural state collected during the recordings? As noted in previous publications (e.g. Ford 1990), behavioural state can affect how whales produce vocalizations, with more aberrant versions of stereotyped calls being produced during periods of socializing. Since behaviour affects how calls are produced, it is possible that call similarity would be influenced by behavioural state. If some families included in the study were only recorded during bouts of intense socializing, the recorded calls may not accurately represent the structure of the stereotyped calls made by those families in other behavioural states more conducive to very stereotyped call production (such as foraging or consistent travel, etc). To truly compare similarity of call structure amongst family groups, shouldn’t behavioural state also be incorporated into the analysis as a variable?

Response: Group behavioral state was noted during the acoustic recordings, but it was not accounted for during this analysis, because it was not necessary. Aberrant calls reported by Ford (1990) are easily discernable from the normal versions of stereotyped calls, and I did not use them for this study.

Line 120: Did the author use a catalog to identify the individuals? Is the catalog citable? It should be cited if so.

Response: I added reference to our catalog ‘The Killer Whales of Eastern Kamchatka’.

Line 139-141: Why 10 calls per call type per group? Is 10 calls enough to capture the variability within the calls adequately? How were the 10 calls per type per group chosen to make up the dataset (were selection criteria based on quality or SNR (or other – like the first ten calls of each type in each recording of each family? The time-warping contour selection is too laborious for a larger sample size?))? Please state reasoning.

Response: Ten call per group were selected as a reasonable number of calls I can obtain from a reasonable number of groups. If I decide to use 20 calls per group, I will be able to get this sample size from only four or five families. It is surprisingly difficult to obtain a sample of even just 10 good-quality calls reliably assigned to a particular family. Killer whales are more vocal in presence of other families – a context that makes call assignment to particular family rather difficult. For example, we have many encounters of two-three families of the same pod with hundreds of great good-quality calls, but we cannot use them, because we do not know which of the families produced them (and as the families are from the same pod, their dialects look identical to me, so I can’t just reveal it by the call structure). We also have a lot of single-family encounters during which the whales were almost totally silent – they usually are not very vocal when only their own family is present. For this reason, I had to limit my sample size to 10 calls per family.

Did any examinations take place that looked at the overall quality of the final selected calls by family group? It seems to me that better quality calls may lead to better measurements in the dynamic time-warping analysis. If some groups had a higher proportion of poor quality calls, this may affect the comparisons. 

Response: All the calls used for the contour-tracking were extremely good quality calls (which is also why I could get so few of them). These calls were initially selected for the comparison of different syllables for my previous paper, and this required very good quality calls with all syllables clearly visible.

The manuscript and interpretability of the results would be improved by including a few more details in the methods about the data set selection process.

 Response: I have added more details on the call selection process.

Line 141: remove ‘each’ from ‘per each family’.

Response: Removed.

Line 142-146: Is the custom MATLAB contour extraction tool previously published? If so, please cite, or provide a few more details on the method: how did the operator determine they’ve selected enough points to adequately describe the contour? If the operator has to do this for each call, this may introduce subjectivity. Could this subjectivity bias the contour shape slightly of each call? If you repeated the point selection and extraction process 10 times from the exact same call clip would the resulting contours be the exactly the same? If not, how much variability exists among these contours? 

Response: The MATLAB script was published for the first time in Filatova et al. 2012. That paper includes more detailed description of the contour extraction process, so I added this reference to the text.

Regarding the subjective variability of contours, I did not assess this, but I expect it to be very low, because the operator can see the resulting contour on the background of the spectrogram and make sure that the contour follows the call precisely. Therefore, the variation does not exceed several Hz, which is a normal error level even when you measure call parameters with AviSoft SASLab or another similar software.

Also, these methods need to be clearer on how the high-frequency and low-frequency contours were treated. Were they selected and extracted independently of each other (so there would be a HFC contour and a LFC contour for each call clip?)? In Line 168-169 the author states: “..and for the contours that consisted of both the components taken together.” Does this mean that the analyst selected points to extract the HFC and LFC as one contour containing features of both? More clarity on the contour extraction process and how the contours for both components were treated is needed to help interpretability and reproducibility.

Response: Yes, the LFC and the HFC contours were extracted independently. When I say “contours that consisted of both the components taken together“ I mean that for this analysis the LFC and HFC contours were concatenated horizontally to form a single contour that included both the LFC and HFC contours – this was necessary to assess the rate of correct classifications when both components were used.

Line 149-150: The author should briefly state why adopting a modified approach was necessary.

Response: The modification was made for our previous analyses and it was described in detail in Filatova et al. 2012: “Because the algorithm of Deecke & Janik (2006) only allows expansion or compression of the time axis by a factor of three, the algorithm cannot be used to compare calls that differ in length by more than a factor of three. In this case, their similarity is considered 0%. This constraint biased the results in comparisons where many short or long contours were present in the repertoire of one population but not the other. To avoid this, we developed an additional algorithm that stretched the shorter contour through interpolation to make it one point longer than a third of the longer contour.” For the current study this modification in fact was not necessary because we compared the calls of the same type, and none of them differed in length by more than a factor of three. We used the modified algorithm only because it was already modified and there was no point to change it back because it does not change anything when the calls have similar length. For this reason, to avoid confusion I have deleted ‘modified’ from the text, because the results did not differ from those that would be obtained by the original algorithm. 

Line 150-155: This description needs to be clearer. It could be made clearer if supplemented by a figure of a contour and an example output from the time-warping algorithm to illustrate what they mean by “smaller frequency value by the larger frequency value at each point” and “a running tab was kept on the similarities of elements making up the curves while adding up these costs…”. More detail into how the cost matrix works would be great. A figure with a flow diagram of how the values from the time-warping get turned into percent similarity, then combined into the cost matrix resulting in a single value.

Response: I did not develop this algorithm and it was described in detail by its authors in Deecke and Janik 2006, which I refer to. The algorithm itself is available online (for example, in the appendix of of Volker Deecke’s thesis). Also, there are millions of illustrations of dynamic time warping process easily available to anyone who can use Google search. Therefore, I believe that describing the details of the algorithm with figures in this particular manuscript would be definitely an overkill.

Line 158-160: This sentence is an exact replica of a sentence in the methods of Deecke and Janik 2006, citation [34] in this manuscript. This sentence must be re-worded to avoid plagiarism.

 Response: Changed.

Line 161-162: the original ARTwarp algorithm was not developed by Deecke and Janik [34], but as Deecke and Janik [34] describe, it was originally developed by Carpenter and Grossberg (1987), and modified for use by Deecke and Janik. 

Response: I changed the wording to “ARTwarp algorithm used by Deecke and Janik”.

Line 176-177: This part of the paper is very interesting and has the potential to provide great insight into how these two call components may be perceived and used by killer whales. However, using relative amplitude of the two components is only valid if the recording system’s frequency response is truly flat. The author previously mentions in the methods that the hydrophones used have a flat response, but provide no information for how that was determined. The detection range comparison compares the relative amplitude across frequencies, and thus, if the recording system does not actually have flat response, the relative contribution of the high and low frequency components will not reflect reality. For example, if the recording system is actually more sensitive in the 5-8kHz band than in the ~1kHz band, the amplitude of the high-frequency component would have been over-represented by assuming a flat response. The author should provide more information on how their recording system’s flat response was determined/verified (end-to-end calibration? hydrophone-only calibration? Manufacturers technical documentation? , etc). 

Response: As stated above, I used the manufacturer’s specifications provided with the hydrophone. The sensitivity of the hydrophone was flat within 3 dB range up to 14 kHz, which is well above the highest of the high frequency component in this study. Three dBs is an acceptable error level given the approximate nature of other parameters, including the averaged killer whale audiogram.

Line 188: The author cites two papers as the source for the audiogram information used. However, the two papers present many different audiograms from many different individual killer whales (and even the composite audiograms presented in the two papers are slightly different). For reproducibility, the author must state which specific audiogram was used.

Response: I used the ‘model’ audiogram from Branstetter et al (2017) for the calculations. I have added this information to the text.

Line 217-221: Figure 2: Figure 2’s axes need to be labelled. Only the y-axis is labelled. Also, the x-axis label has a spelling error.

Response: I have fixed the axes labels.

Line 242: “All these calls…” This should be more specific, such as: “All of the calls without detectable high-frequency components…”

Response: Changed.

Line 271: Discrimination of a caller’s orientation, not ‘direction’. The direction to a caller is not related to the high-frequency component’s directionality (their hearing is directional and thus can infer direction to a caller inherently). The high-frequency component gives information about the orientation of the caller relative to the receiver. 

Response: Changed to ‘orientation’.

Line 276-278: Direction of movement does not necessarily equate to the direction of calling. The whale could temporarily face a certain direction to emit a call, then continue on it’s intended path. Also, the high-frequency component is not necessary for determining direction of travel. The caller only needs to emit two omnidirectional sounds of any type while travelling and the listener can discern direction of travel. In this context, the higher frequency component’s directionality can only provide the caller’s orientation. And the orientation information is quite ambiguous, especially considering the 4D environment the whales live in and the number of possible orientations that are possible for the same amplitude ratio of the high to low frequency component. And on top of this, the whales make other types of directional sounds, so the HFC isn’t the only vocalization that can elude to orientation of the caller (I think echolocation is likely a better indication of orientation because it is made very frequently and consistently and the returning echoes from features can also be used…). 

This is all to say that the author can strengthen their argument that the HFC is not solely used for orientation/group coordination using the above points. Also, the discussion can be improved by speaking to the other potential reasons for the directionality in the high-frequency component. The directionality of the HFC can be more useful than for just providing orientation information: the directionality of the component would allow a caller to emit family/clan membership information towards specific whales or groups within an aggregation selectively (just as a human in a noisy room would yell (everyone in the room would hear), but ALSO make eye contact (directional component) with a person on the other side of the room if they were the intended target of such communication). The directionality can also help increase the SNR in the direction of a receiver in noisy conditions to be heard better in noise – not just the simple presence of the high frequency component, because that can be masked quite easily in some situations, but the ability to create a higher SNR by directing the call towards other members of the group in noisy conditions). 

I think the directionality aspect of the HFC is a very important functionality of the HFC and should be discussed more in the discussion (because it ultimately helps out the author’s arguments and final conclusions). Being able to provide family membership more reliably is fine, but the ability to provide family membership in certain directions is a very useful tool for a very social species (including in noisy conditions). 

Response: Thank you! I have added more discussion of the directionality aspects, as well as the argument that other sounds are directional too.

Line 282-284: I don’t think this is necessarily true. The structure of a tone (whether it is frequency modulated or not) can also affect its detectability in different noise conditions. There are types of background noise where a non-frequency modulated tone would be more detectable than a modulated one, but there are also situations where a modulated tone would be more detectable than a non-modulated one. This argument needs to be strengthened.

Response: I have shifted the focus of this statement to address the reviewer’s comment, now it reads “there would be no need to vary its shape across families: the same contour for all families would work equally well”.

Line 290-292: I think this sentence seems a little abrupt, as these uses of biphonation haven't been brought up in the paper before, and also not in relation to killer whales - the abruptness hides the intended purpose of the sentence. I suggest changing the sentence to make it less abrupt, and capture the attempted purpose of the sentence: "Additional hypotheses for the function of biphonation in other species have included: increasing unpredictability and providing an indication of physical condition [19]. However, these are likely not appropriate considerations as possible functions of biphonation in killer whales. These hypotheses suggest that the caller can choose when to include or not include a biphonic component in each call it makes. However, calls that comprise killer whale dialects are stereotyped and ..." Or something like that. 

Response: Thank you! I have included this in the text.

Line 294: The use of the term ‘mandatory’ is slightly misleading. Yes, in the specific call types that contain HFC’s, the HFC component is mandatory (the whale does not choose to whether include or remove the component), but the way the sentence is written it could be interpreted that the HFC is mandatory in any call the whale makes.

Response: To clarify this, I added “if a call is biphonic” to the text.

Line 297: “probably on the family level…” Why only at the family level? In this manuscipt, the author only examined the frequency-time component of the call components, but more information is provided in a vocalization, such as it’s timbre? Could this not carry individual information in the HFC/LFC? There needs to be a citation for why the author believes only family level is possible in killer whales – there is arguably strong pressure for individual recognition in resident killer whales – such as in kin-directed prey-sharing, Wright et al. 2017.

Response: I strongly agree that killer whales can recognise their family members individually by their calls, but this sentence only speaks about the high-frequency component “Most likely, the higher-frequency component of killer whale biphonic calls functions as an alternative contour that bears information on the caller’s identity (probably on the family level, in contrast to individual level in dholes and penguins).” It does not imply that killer whales can’t recognize each other on the family level.

Line 312: ‘shifted’ may not be the appropriate word. “concentrated” or “is more pronounced in lower frequencies”, etc. The noise isn’t shifted to lower frequencies, it is just the nature of most anthropogenic ambient noise. 

Response: Changed to “more pronounced in lower frequencies”, thank you!

Line 319-320: The author should also consider addressing this argument in relation to Viers et al 2016: “Ship noise extends to frequencies used for echolocation by endangered killer whales”. It should be noted that, at times, even the HFC can be masked by ship/anthropogenic noise. But the author can also add another advantage of the directionality of the HFC in this context. If noise levels are high, even to the point of starting to mask the HFC, the caller can point in the direction of the receiver while calling to increase the HFC’s SNR. This all adds to the potential function of this component.

Response: I have added the reference to Viers et al 2016 and the possible additional function of directionality.

Line 324-325: “…better hearing sensitivity of killer whales to higher frequencies.” Cite [35, 36] here.

Response: Done.

Line 338: This is a very cool possibility, but I wonder if it is meaningful biologically? The heterodyne frequencies only show up when call is good quality (high SNR or close), and in this sense, it may not be that effective when in noise because to resolve it the whales need to be relative close to one another, and if they are relatively close, they will already be able to discern the low frequency component? 

Response: This is probably true in most situations, but still I can imagine situation when there is very intensive noise on low frequencies that masks the lower-frequency component but the higher-frequency component and the heterodynes are audible far enough on the caller’s axis. Anyway, it is just a speculation, but I think an interesting one.

Line 357: “pods” should be replaced by “groups” or “social units” for clarity.

Response: Changed to ‘social units’.

Line 362: “This process of cultural change is called cultural evolution.” The author is not defining what cultural evolution is for the first time in this paper. Please cite.

Response: I added references to Mundinger (1980) and Lumsden & Wilson (1985).

Line 363: use of ‘pods’ needs to be defined or changed to ‘groups’.

Response: Changed to ‘social units’.

Line 368: “…probability that both components would converge in two pods is very low.” The probability is likely relative to whether the two ‘pods’ are members of the same or different populations. And if in part of the same population, relative to the overall population size and the frequency that different groups in that population encounter one another.

Response: No, I mean random convergence here – the one that occurs just because the number of contour shapes is limited, and therefore there is always a probability that in the process of cultural change two contours can randomly become more similar. 

Line 371-377: This paragraph would benefit from re-wording to provide better clarity. 

Response: I have re-worded this paragraph.

Line 402-405: I think these conclusions can benefit from a more thorough discussion of the advantages of directionality (as mentioned previously in these comments). 

Response: I have added the statement about the directionality here.

Supplemental Information:

S1: I noticed that the matrices for the K5 and K7 call types are different sizes. Why are the K5 datasets 150x150 whereas the K7 datasets are 140x140? In the methods the author describes data collection as: 14 families, 10 calls of each type per family, so shouldn’t they both be 140x140? I may have misunderstood something? If not, then the author should address this in the methods.

Response: Sorry, I should have clarified it in the methods. One of the families had two distinctive subtypes of K5 calls, and I used them both for the analysis, so from this family I selected 20 calls (10 from one subtype and 10 from another). I have added these details to the methods. 

Reviewer #4: Overall this is a valuable study that advances our understanding of this acoustic phenomenon in an important way. The conclusion that “the main function of the higher-frequency component is to duplicate and/or complement the social identity marking when the lower-frequency component is masked by noise or accidentally appears similar in unrelated social units.” Is well supported by the analyses and so I recommend the paper be accepted subject to some changes needed in interpretation and presentation.

My biggest concern is that the Title and second primary conclusion is not adequately justified as ‘independent cultural change’ has not been shown- that can only be done using a temporal study showing how the features have changed over time. The pattern found could just be due to natural variability within otherwise fixed contours – or contours that slowly evolve but not independently. New title could be: Independent acoustic variation of higher-frequency components can facilitate call recognition in killer whales

Response: I have changed ‘cultural change’ to ‘acoustic variation’ in the title and the abstract. In the Discussion, the independent cultural change is mentioned in the context of other papers that have demonstrated the change of call features over time, so I believe it is justified there.

The manuscript is important without this unsupported claim. You should propose specific future work by which the question of whether or not these call types truly evolve independently is tested more directly and robustly.

A second overall concern is that the write-up doesn’t cite all of the most relevant work in the field. It is important to cite and discuss those published studies that most closely relate to your current work.

 Response: I have added the suggested citations.

Minor suggestions:

Ln 39 - suggest you delete ‘excellent’. The cited study shows rather poor copies of sounds made by one captive killer whale.

 Response: Deleted.

Ln 40 add ‘call repertoire’ before dialect to be more precise.

Response: Added.

Ln 46-‘evolves’

Response: Changed.

Ln 49 – no ‘the’ before ‘social’

Response: Deleted.

Ln 55- A paper by Deecke The structure of stereotyped calls reflects kinship and social affiliation in resident killer whales (Orcinus orca) - seems important to cite here.

 Response: Added.

Ln 59 or 62. A paper by Miller Caller sex and orientation influence spectral characteristics of “two-voice” stereotyped calls produced by free-ranging killer whales -seems important to cite here.

Response: Added.

Ln 149-it is strange to read ‘we’ for a single authored paper.

Response: I have changed ‘we’ to ‘I’ where appropriate.

Ln 158 and Discussion: One highly relevant paper that isn’t cited here looked at error rates of classification for LFC versus HFC: “The influence of social affiliation on individual vocal signatures of northern resident killer whales (Orcinus orca)” It appears that very similar methods and results were found, though on a slightly different social scale.

Response: Thank you for drawing my attention to this paper! I’ve read it before, but I have forgotten that it had compared the LFC and the HFC in the similar way to what I did. Unfortunately, that paper is very brief and does not provide enough details to compare our results. Nevertheless, I mentioned it in the Discussion.

Methods: consider the possibility that the ‘same family’ results may have been influenced by some of the calls being produced by the same individual.

Response: It is definitely possible, but there is no way around it because many of the families had less than 10 whales, so there is no way to select 10 calls from such family without some degree of pseudo-replication.

Ln 225 and 226 – add ‘pairs of’ before ‘calls’

Response: Added.

Table1and Figure 2 – what are the units here? This value seems to come from a black box, so make an effort to convince the reader that it is a valid indicator of the similarity.

Response: Technically, these are percent. The dynamic time warping gives the result as % similarity, and to calculate distances I subtracted the similarity values from 100%. I added these details to the Methods.

Table 1- this is a mean of the distances averaged across a lot of pairs of calls, right? If so, state that this is a mean value, and add the standard deviation and sample size to the table.

 I have added the SD and stated that the table has mean +-SDs in the title. However, I do not think it makes sense to add sample size. The sample size here is the number of pairs, which is much larger than the number of calls. Adding the number of pairs looks like inflating the real sample size, while adding the number of calls would be confusing because the values were calculated from the number of pairs.

Table 1 title is confusing- State that is the distance between pairs of calls.

 Response: Done.

Figure 2-consider to use smaller symbols and open face symbols to better show the data underneath other data points.

Response: I have made the symbols twice smaller. Open face symbols are ugly; I think the semi-transparency is good enough to show the points underneath.

Figure 2 – add an x-axis label. Correct top panel y-axis label.

 Response: Done.

Ln 240-246: Did you ever see cases when only the higher component was visible?

Response: No. I would expect this to happen when the lower component is masked by intensive low-frequency ship noise, but we usually don’t do recordings when there are loud ships nearby.

Discussion:

The difference between a signal and the audiogram sensitivity at that frequency is commonly known as the ‘sensation level’. This is a dictionary definition, and could be useful for your paper.

Response: Thank you very much! I have used it where appropriate.

Ln 309 – how much more quickly does the higher frequency component attenuate than the lower frequency component-in dB/km? Is that difference enough to overcome the higher sensation level of the higher component with distance? (personally I think the difference is very small, but it is worth to show you looked at this).

Response: I checked this before and frequency-dependent absorption differences were indeed rather low compared to killer whale sensitivity differences to higher frequencies. For example, the difference between 1 and 10 kHz is about 0.7 dB per km. Given that normally we record killer whales within few kilometers, the difference would be only 1-2 dBs. I am not sure I should mention this in the text, because anyone who knows about the frequency-dependent absorption would also probably realize that its influence is low, like you do.

LN 331-333: Again here it’s relevant to cite Miller et al Caller sex and orientation influence spectral characteristics of “two-voice” stereotyped calls produced by free-ranging killer whales -- as it supports the presence of heterodyne frequencies in the two component calls from a totally different population of killer whales.

Response: Reference added.

Ln 336-338. I’d suggest caution here regarding your point on evolution of call types. It is clear that we don’t know the actual mechanism by which these contours are produced, which limits our ability to be certain that each component can be modified independent of the other. There may be aspects of sound production that fundamentally limit the flexibility of one component to change without affecting the other.

Response: Not sure which lines this comment referred to, because lines 336-338 were actually about heterodynes, not about evolution. Anyway, I agree that we don’t know the actual mechanism, but the fact that their variation is independent directly follows from the results. If the flexibility of one component was limited, then the distances between the pair of its contours would be much lower than of the other non-limited component, but the results show that the distance variation of both components is at about the same scale.

Ln 399- all of this evidence is consistent with independent evolution of the two components, but doesn’t demonstrate it directly and rigorously. Alternative drivers of the patterns you found remain possible. Stronger to here to indicate future temporal analyses that would be able to tackle the question more directly.

Response: I have added these two sentences here: “However, despite all of this evidence is consistent with independent evolution of the two components, it does not demonstrate it directly and rigorously. In future, analysis of the component variation over time is necessary to exclude the alternative drivers of the observed patterns.”

Acknowledgements reads odd with ‘We’ if this is a single author work.

Response: Changed to “I”.

Overall, very good work.

 Response: Thank you!

---

## [Decision Letter · Decision Letter 1]

24 Jun 2020

PONE-D-20-05337R1

Independent acoustic variation of the higher- and lower-frequency components of biphonic calls can facilitate call recognition and social affiliation in killer whales

PLOS ONE

Dear Dr. Filatova,

Thank you for submitting your manuscript to PLOS ONE. After careful consideration, we feel that it has merit but does not fully meet PLOS ONE’s publication criteria as it currently stands. Therefore, we invite you to submit a revised version of the manuscript that addresses the points raised during the review process.

The authors have done a very good job of addressing the original reviews. Two of the original reviewers (2 and 4) reviewed the revised manuscript, and Reviewer 4 has a few minor suggestions that could further improve the manuscript. Please respond to these additional comments. Following this, the manuscript should be acceptable for publication.

We look forward to receiving your revised manuscript.

Kind regards,

William David Halliday, Ph.D.

Academic Editor

PLOS ONE

Additional Editor Comments (if provided):

The authors have done a very good job of addressing the original reviews. Two of the original reviewers (2 and 4) reviewed the revised manuscript, and Reviewer 4 has a few minor suggestions that could further improve the manuscript. Please respond to these additional comments. Following this, the manuscript should be acceptable for publication.

Reviewers' comments:

Reviewer's Responses to Questions

**Comments to the Author**

1. If the authors have adequately addressed your comments raised in a previous round of review and you feel that this manuscript is now acceptable for publication, you may indicate that here to bypass the “Comments to the Author” section, enter your conflict of interest statement in the “Confidential to Editor” section, and submit your "Accept" recommendation.

Reviewer #2: All comments have been addressed

Reviewer #4: (No Response)

2. Is the manuscript technically sound, and do the data support the conclusions?

Reviewer #2: Yes

Reviewer #4: Yes

3. Has the statistical analysis been performed appropriately and rigorously? 

Reviewer #2: Yes

Reviewer #4: Yes

4. Have the authors made all data underlying the findings in their manuscript fully available?

Reviewer #2: Yes

Reviewer #4: Yes

5. Is the manuscript presented in an intelligible fashion and written in standard English?

Reviewer #2: Yes

Reviewer #4: Yes

6. Review Comments to the Author

Reviewer #2: (No Response)

Reviewer #4: I have a few minor comments that could improve the manuscript further.

Ln 26 – could change ‘perceived amplitude’ to ‘sensation level’ to use standard terminology

Ln 273 – this section has ‘over distance’ in the title, but no consideration of propagation effects or distance over which they might be used are mentioned here. Consider to change the title of this section. Alternatively, you could mention that frequencies of the HFC will absorb roughly 1dB/km more than the LFC and consider the consequences of this difference on the typical detection ranges indicated in Miller et al., 2006 – At 10km distance, the HFC will have lost 10dB more than the LFC.

Ln 279 – the HFC may also not be detectable if the whales were oriented away

Ln 338: could change ‘mandatory’ to ‘obligatory’

Ln 457: change ‘the main’ to ‘a likely’ – until we are able to test these different ideas using playback experiments all of the proposed functions remain somewhat speculative.

Ln 461: the directionality means that this will not be audible as far to receivers that aren’t ahead of the signaller. This is a potential drawback to its use as a general family indicator, so it may not really be ‘especially suitable’. The combination of both components does provide a redundancy of information that is beneficial for these animals to maintain contact with preferred group members.

7. PLOS authors have the option to publish the peer review history of their article (what does this mean?). If published, this will include your full peer review and any attached files.

Reviewer #2: No

Reviewer #4: No

---

## [Author Response · Author response to Decision Letter 1]

11 Jul 2020

Reviewer #4: I have a few minor comments that could improve the manuscript further.

Ln 26 – could change ‘perceived amplitude’ to ‘sensation level’ to use standard terminology

Response: Changed.

Ln 273 – this section has ‘over distance’ in the title, but no consideration of propagation effects or distance over which they might be used are mentioned here. Consider to change the title of this section. Alternatively, you could mention that frequencies of the HFC will absorb roughly 1dB/km more than the LFC and consider the consequences of this difference on the typical detection ranges indicated in Miller et al., 2006 – At 10km distance, the HFC will have lost 10dB more than the LFC.

Response: I have added the explanation about the frequency-dependent absorption and the reference to Miller (2006).

Ln 279 – the HFC may also not be detectable if the whales were oriented away

Response: If the whale is close, the HFC will be detectable even if it is oriented away (I have seen it many times when whales were passing our boat, calling first towards, and then away from the hydrophone – the HFC was weaker in the latter case, but still detectable). Therefore, the orientation away decreases the received level of the HFC, but it becomes undetectable due to transmission loss, which is already mentioned in the text.

Ln 338: could change ‘mandatory’ to ‘obligatory’

Response: Changed.

Ln 457: change ‘the main’ to ‘a likely’ – until we are able to test these different ideas using playback experiments all of the proposed functions remain somewhat speculative.

Response: Changed.

Ln 461: the directionality means that this will not be audible as far to receivers that aren’t ahead of the signaller. This is a potential drawback to its use as a general family indicator, so it may not really be ‘especially suitable’. The combination of both components does provide a redundancy of information that is beneficial for these animals to maintain contact with preferred group members.

Response: I have deleted the statement that the directionality makes these calls ‘especially suitable’ and added the statement that combination of both components provides a redundancy that is beneficial in their environment.

---

## [Editor Report · Decision Letter 2]

14 Jul 2020

Independent acoustic variation of the higher- and lower-frequency components of biphonic calls can facilitate call recognition and social affiliation in killer whales

PONE-D-20-05337R2

Dear Dr. Filatova,

We’re pleased to inform you that your manuscript has been judged scientifically suitable for publication and will be formally accepted for publication once it meets all outstanding technical requirements.

Kind regards,

William David Halliday, Ph.D.

Academic Editor

PLOS ONE

Additional Editor Comments (optional):

This manuscript can now be accepted. I did find one very minor spelling error on line 284: "loose" should be "lose". This can be changed prior to production.
---

## [Editor Report · Acceptance letter]

17 Jul 2020

PONE-D-20-05337R2 

Independent acoustic variation of the higher- and lower-frequency components of biphonic calls can facilitate call recognition and social affiliation in killer whales 

Dear Dr. Filatova:

I'm pleased to inform you that your manuscript has been deemed suitable for publication in PLOS ONE. Congratulations! Your manuscript is now with our production department. 

Kind regards, 

on behalf of

Dr. William David Halliday 

Academic Editor

PLOS ONE